# The Potential Benefits of Acute Aronia Juice Supplementation on Physical Activity Induced Alterations of the Serum Protein Profiles in Recreational Runners: A Pilot Study [note 1]

**DOI:** 10.3390/healthcare12131276

**Published:** 2024-06-26

**Authors:** Tamara Uzelac, Marija Takić, Vuk Stevanović, Nevena Vidović, Ana Pantović, Petar Jovanović, Vesna Jovanović

**Affiliations:** 1Department of Biochemistry and Centre of Excellence for Molecular Food Sciences, Faculty of Chemistry, University of Belgrade, Studentski trg 12-16, 11000 Belgrade, Serbia; tamarauzelac31@gmail.com; 2Institute for Medical Research, National Institute of Republic of Serbia, Group for Nutrition and Metabolism, Centre of Research Excellence in Nutrition and Metabolism, University of Belgrade, Tadeuša Košćuškog 1, 11000 Belgrade, Serbia; vuk.stevanović@imi.bg.ac.rs (V.S.); nevenakardum@gmail.com (N.V.); jelenkovicana5@gmail.com (A.P.); petarjovanovichfbu@gmail.com (P.J.)

**Keywords:** aronia, half-marathon, proteinuria, protein profiles, recreational runners

## Abstract

Intensive physical activity (PA) can lead to proteinuria and, consequently, serum protein profiles in athletes. Therefore, the aim of this study was to investigate the effects of acute aronia juice consumption before a simulated half-marathon race on serum protein profiles in recreational runners. The pilot study was designed as a single-blind, placebo-controlled, crossover study, with 10 male participants who consumed aronia juice (containing 1.3 g polyphenols) or placebo before the race. The blood levels of total proteins, albumin, the non-albumin fractions gamma, beta, alpha2 and alpha1, as well as renal function parameters, were determined before and 15 min, 1 h and 24 h after the race. The significant changes in urea, creatinine and uric acid levels were noticed at selected time points in both groups. In the placebo group, a significant decrease in total proteins (*p* < 0.05) was observed 24 h after the race, along with an increase in gamma fraction abundance (*p* < 0.05). In addition, urea and uric acid levels returned to baseline only in the aronia group 24 h after the race. Thus, according to the results obtained, acute aronia juice supplementation before intensive PA could influence the transient change in renal function and PA-induced protein loss in recreational runners.

## 1. Introduction

The importance and beneficial effects of leisure-time physical activity (PA) on human health have been well studied [1]. It is defined as a “PA performed during exercise, recreation or any time other than those associated with one’s regular occupation, housework, or transportation”. Due to the growing awareness of peoples’ health and wellbeing, the trend of recreational PA has been increasing in most countries in the last two decades [2]. The most popular PAs are walking and running, especially long distance running (half-marathon, marathon and ultra-marathon). Besides the investigation of beneficial effects of PA, the knowledge about negative effects like oxidative stress, dehydration, proteinuria, muscle damage and acute kidney injury, which occur during PA, has also been significantly enhanced [3]. A moderate or an intensive PA can lead to changes in many blood biochemical and hematological parameters (such as level of glucose, total proteins, creatinine, urea, uric acid, the activity of different enzymes, the content or activity of different components of antioxidative system, the count of erythrocytes, leukocytes, thrombocytes, and/or hemoglobin concentration) [4]. If these negative changes usually induced by repetitive long-term PA are not appropriately treated, they can progress further [3]. Acute kidney injury is a relatively common complication caused by intensive PA [3]. Repetitive episodes of acute kidney injury may lead to chronic kidney disease [5], which can be an issue for some recreational runners who present unhealthy behaviors, such as running a marathon every week [6]. The potential beneficial effects of supplementation with vitamins, polyphenols, and amino acids in the prevention of negative effects of intensive PA have been recently considered [7].

Dehydration, proteinuria, and muscle damage caused by moderate or intensive PA can change the content of total proteins, human serum albumin (HSA) and non-albumin proteins in the blood [8,9,10]. When the amount of excreted water exceeds the water replacement, then dehydration occurs, and it is accompanied by increases in concentrations of different blood components including proteins. Dehydration is the only known cause of hyperalbuminemia in humans [11,12]. Muscle damage induced by PA leads to a release of proteins from muscle cells. In trained athletes, muscle damage is often mild, and leads to transient increases in markers of muscle damage like creatine kinase and lactate dehydrogenase in the circulation [10]. An acute kidney injury is a relatively common complication after intensive or moderate PA, which has an impact on the blood and urine content of different biochemical parameters. Different parameters are recommended as the markers of kidney injury, such as the serum level of creatinine, urea and uric acid or albuminuria, according to Kidney Disease Improving Global Outcomes criteria [13]. Albuminuria is the best known urinary marker of kidney dysfunction, caused by excess excretion of serum albumin into urine. Beside HSA, other proteins from the circulation can be selectively excreted by the kidney during PA, which can lead to a decrease of total protein content and changes of their profiles. 

In PA, proteinuria is considered as a benign phenomenon, because it is not related to any structural changes in the kidney [3]. This transient functional change of the kidney is caused by different mechanisms [14]. The induction of renin-angiotensin 2 system, catecholamine synthesis induced by sympathetic system, and the lactate-induced change of protein structure and renal membrane permeability are common mechanisms by which higher renal permeability and transient proteinuria during intensive PA are explained [15].

Aronia (Aronia melanocarpa (Michx.) Elliot) is a shrub cultivated worldwide. It gained the greatest popularity in the northern, eastern (including Serbia), and central countries of Europe. Aronia grows in a wide variety of conditions, as it is highly adaptable, giving it a high yield. It can grow in mild climatic conditions, but also at extremely low temperatures [16]. The composition of aronia fruits depends on various factors, such as maturity, climatic, and environmental conditions. Aronia fruits are rich sources of phenolic compounds, including anthocyanins, phenolic acids, proanthocyanidins, and other subclasses. Polyphenols are compounds that govern the high bioactivity of aronia berries [17]. So far, in various physiological and pathological conditions, the numerous beneficial effects of aronia juice supplementation have been confirmed [18]. A recent study reported that consumption of anthocyanins from aronia showed beneficial effects on renal function in animals [19]. In addition, another study showed that aronia polyphenols mainly exerted their effect on hypertension by influencing the renin-angiotensin system of the kidneys [20]. Although several recent studies have investigated the effects of aronia juice supplementation on sport-related outcomes in athletes [21,22,23], to the best of our knowledge, its effects on PA-induced proteinuria have not yet been investigated [24].

Running is the most popular recreational sport, with millions of enthusiasts worldwide [25,26]. To the best of our knowledge, there is no literature data about the possible effects of the long-lasting PA on serum protein profiles, especially in recreational runners during repetitive moderate and intensive PA followed by persistent proteinuria. Besides this, the joint influence of PA and acute aronia juice consumption on the serum protein profiles has not been investigated so far. The main goal of this pilot single-blind crossover was to examine the effect of aronia juice supplementation versus placebo on serum protein profiles in male recreational runners at selected time points before and after simulated half-marathon races. To investigate the possible effect of proteinuria, and dehydration caused by PA on the serum proteins profiles, aside from some biochemical parameters such as determination of creatinine, urea and uric acid, serum total proteins, HSA and non-albumin proteins content were determined. Additionally, to obtain insight into PA-induced dehydration and proteinuria effects on the serum protein profiles, the abundance (% and g/L) of five main serum protein fractions (gamma, beta, alpha2, alpha1 and HSA), as well as the ratios of three of the most abundant fractions (gamma, beta and HSA), were also determined in all the selected time points after their separation by native electrophoresis.

## 2. Materials and Methods

### 2.1. Subject and Design

The study was conducted according to the Declaration of Helsinki, and the protocol was approved by the Ethics Committee of the Ethical Committee of the Faculty of Sport and Physical Education, University of Belgrade, Serbia (ref. no. 02-1072/18-1). Informed consent was sought from all participants. A pilot, single-blind crossover placebo-controlled study (Figure 1), that was described in detail elsewhere [27], was designed to determine the effects of PA and/or acute aronia juice consumption before simulated half-marathon races on protein profiles in serum recreational runners. Briefly, 10 participants, (apparently healthy recreational male runners who are members of the Belgrade Urban Running Team) with 4 practices per week (approximately 40 km of weekly running distance), consumed 200 mL of commercially available aronia juice or a placebo juice shortly after breakfast, 1 h before the simulated half-marathon races. The wash-out period between two simulated half-marathon races that each participant ran was 7 days (Figure 1). The average times needed to end the race in aronia and placebo group were 121 and 112 min, respectively (Stevanović, 2019), and these times could indirectly give information about the intensity of physical activity during the race and physical performance of runners. According to the average times required to complete the simulated half-marathon races, as well as the ages of runners in this study, their intensity of physical activity was at the novice level. (https://runninglevel.com/running-times/half-marathon-times, accessed on 10 June 2023). The commercially available aronia (black chokeberry) Nero cultivar juice was donated from Rheapharm d.o.o., Belgrade, Serbia. Briefly, the juice was made by mechanical processing, including homogenization and pressing, from ripe, carefully selected fruits of *Aronia melanocarpa*, harvested in August/September from a plantation field on the mountain Suvobor (750 m a.s.l.), Serbia. The meals were identical according to calories and macronutrient compositions. The placebo juice had the same nutrient content, taste, color, smell, and texture as aronia juice, but was without polyphenols [28]. Total polyphenol amount was 1.3 g in 200 mL of aronia juice, expressed as gallic acid equivalents [27]. The juice from the same batch was provided to all the participants. The juice/placebo was given shortly after breakfast because of its strong astringent taste and to prevent it from causing an upset stomach. Based on previous studies that investigated the acute effects of polyphenol-rich sources on exercise-induced oxidative stress, we decided to give aronia/placebo juice to the participants before exercise [29,30,31]. In addition, the timing of aronia consumption was chosen, so that peak anthocyanin and total polyphenols concentration would occur in the blood during the exercise [32]. The dose of the juice approximately matched the polyphenol content in other polyphenol-rich fruit juices that were previously investigated in recreational and elite runners [33,34]. The placebo juice was originally developed as a control beverage, to be used in intervention studies with aronia juice. More precisely, three formulations were prepared by mixing various nutrients, matching the nutrient composition of aronia juice, in water, with the addition of artificial colors and flavors. As previously published, the similarity of these formulations to aronia juice was assessed by six food panelists so that the formulation was chosen [23,24,25,26,27,28].

### 2.2. Blood Sampling

After an overnight fast before breakfast, a venous blood sample was collected before (T0), and 15 min (T1), 1 h (T2), and 24 h (T3) after each race into serum sample tubes. One tube was collected per participant at each time point, a total of four tubes per race. Prior to each blood sampling, participants rested for at least 10 min. Blood samples were allowed to clot at room temperature, and sera were separated by centrifugation (4000× *g* for 10 min at 10 °C), frozen and stored (at −70 °C) before further analyses.

### 2.3. Determination of Total Serum Proteins, HSA, and Non-Albumin Proteins

The concentration of total serum proteins was determined by the biuret method [35]. The bromocresol green method was used for the quantification of serum HSA concentration [36]. The calibration curves, constructed from HSA concentrations ranging from 1–100 or 1–80 g/L, were used for the determination of protein or HSA concentration, respectively. The obtained values were expressed in g/L. The concentration of non-albumin proteins in serum was determined according to Equation (1).
Non-albumin proteins (g/L) = total proteins (g/L) − HSA (g/L) (1)

### 2.4. Determination of Serum Protein Profiles

Native polyacrylamide gel electrophoresis (PAGE) was performed according to the manufacturer’s recommendations, using a Hoefer SE 260 electrophoretic unit (San Francisco, CA, USA) for the separation of five main serum protein fractions (gamma, beta, alpha2, alpha1 and HSA) [37]. Before electrophoresis, the concentration of proteins in the serum was determined by the biuret method [35]. The experiment was performed in duplicate. In the first experiment, 5 µg of serum proteins was applied per line, while in the second 7 µg of proteins was applied on 9 % polyacrylamide gel for the better separation of single serum protein fractions. The proteins in the gel were stained by CBB. The percentage abundances of protein fractions were determined by a densitometric analysis of electropherograms, obtained after the scanning of the gels using ImageJ.

### 2.5. Determination of Urea, Creatinine and Uric Acid

The concentrations urea, creatinine and uric acid in serum were determined using the clinical chemistry analyzer Cobas c111 and reagent kits (Roche Diagnostics, Basel, Switzerland), as recommended by the manufacturer. Briefly, creatinine and uric acid were determined by enzymatic colorimetric methods. Creatinine was converted into glycine, formaldehyde and hydrogen peroxide using creatininase, creatinase, and sarcosine oxidase, while uric acid was converted into allantoin and hydrogen peroxide using uricase. In the next step, the liberated hydrogen peroxide was converted into quinone-imine chromogen by the action of peroxidase, whose color intensity was directly proportional to the creatinine or uric acid concentration (Roche Diagnostics, Basel, Switzerland). Urea was measured by the kinetic enzymatic assay, in which urea was hydrolyzed by urease into carbonate and ammonium. In the next step, these products were converted into L-glutamate in the presence of glutamate dehydrogenase and coenzyme NADH. The rate of the decrease in the concentration of NADH was directly proportional to the concentration of urea in the sample [38].

### 2.6. Statistical Analysis

Each assay was performed in triplicate. The results are expressed as mean ± SD. The statistical significance of obtained differences between groups, and different time points in each group, were tested using paired *t*-test. All statistical analysis and graphical representations of data were performed using the Origin 9.0 statistical program. Values *p* < 0.05 were considered as significant.

## 3. Results

### 3.1. Baseline Characteristics of the Study Participants

Ten healthy recreational male runners between 25 and 35 years were participated in this study. The mean values of main characteristics of participants were 30.8 ± 2.3 years, height 185.3 ± 7.4 cm, body mass 84.3 ± 13.5 kg and body mass index 24.6 kg/m^2^. Additionally, the baseline biochemical (glucose, triglycerides, total, HDL and LDL cholesterol), and the hematological parameters of the study participants were within the normal reference ranges [27].

### 3.2. Biochemical Parameters of Renal Function

Serum levels of creatinine, urea and uric acid, as the recommended markers of kidney injury, were determined before (T0) and after race (T1–T3). In both groups, the concentrations of urea, creatinine and uric acid were significantly higher in time points T1 and T2 compared to T0 (Table 1). In the placebo group, the levels of urea and uric acid were significantly higher (*p* ˂ 0.01) 24 h after race, while they returned to baseline values in the aronia group. Analyzing normalized values, a significant treatment effect was observed for urea (*p* < 0.05) and uric acid (*p* < 0.01) 24 h and 15 min after the race, respectively.

The results obtained suggest the presence of transient, functional renal damage, as creatinine levels were elevated immediately after the race and then returned almost to baseline levels in both groups.

### 3.3. The Content of Total Proteins, HSA, and Non-Albumin Proteins

Tracking serum total protein, HSA, and non-albumin protein levels can provide insight into processes such as dehydration and proteinuria during PA. Dehydration cannot lead to changes in the ratio of HSA to non-albumin proteins, whereas proteinuria can. In both groups, the concentrations of total serum proteins increased at time points T1 and T2, but these increases reached statistical significance (*p* ˂ 0.05) only in the aronia juice group (Table 2). A significant (*p* < 0.05) increase in HSA concentration was only observed at time point T2 in the aronia group. In the placebo group, a lower concentration of total proteins (*p* < 0.05) was observed 24 h after the race compared to the corresponding baseline values (T0), while the decrease in HSA and non-albumin proteins did not reach statistical significance. In addition, no significant treatment effect was observed for total proteins, HSA, and non-albumin proteins (*p* > 0.05). The transient increase in all serum protein fractions 15 min and 1 h after the race in both groups compared to T0 values indicates that dehydration occurred, but different trends in changes in HSA and non-albumin protein fractions from T1 to T2 indicate that proteinuria was also present.

To confirm that two intensive Pas, with a short period of time in between, can affect the levels of serum protein fractions due to proteinuria, their levels were plotted 24 h and 7 days after the first race, regardless of treatment (Figure 2). It seems that the 7-day period between two races was not sufficient for the complete recovery of total proteins, as the mean concentration values of total proteins, HSA and non-albumin proteins before the 2nd race were 3.3 (*p* < 0.01), 2.5 and 0.8 g/L lower than the mean values before the 1st race, respectively. It can also be seen that the mean values of HSA and non-albumin proteins were at the same level 24 h and 7 days after the first race. This indicates that proteinuria influences the values of both fractions.

### 3.4. Serum Protein Profiles

It is known that non-albumin proteins can be separated into four different fractions by native electrophoresis: alpha1 (α1), alpha2 (α2), beta (ß) and gamma (γ) [37]. The determination of serum protein profiles, together with the determination of changes in total protein concentration, allowed us to gain some insights and conclusions on how dehydration and proteinuria due to PA might affect the changes in protein levels. Protein profiles were presented as a percentage abundance of HSA and γ, β, α1 and α2, obtained after the densitometric analysis of gels following the separation of serum proteins by native electrophoresis (Table 3 and Appendix A).

According to the results presented in Table 3, the abundances of γ and β obtained at time point T1 were almost unchanged compared to the baseline values, while the trends of changes in the abundances of α2, α1 and HSA were different in both groups. At time points T2 and T3, an increase in the percentage of the γ fraction was observed in both groups compared to baseline values, but this was only statistically significant (*p* ˂ 0.05) in the placebo group. Looking at the non-statistically significant changes in T2 and T3, α1 fraction abundance decreases in the placebo group in T3. 

According to these results, HSA and α1 could contribute to a significant decrease in total serum proteins 24 h after the race in the placebo group (Table 2). In addition, the decrease in these protein fractions is probably jointly responsible for the decrease in total proteins 24 h and 7 days after the first race (Figure 2). The decrease in total proteins is likely due to proteinuria, as both total protein levels and protein profiles were altered.

In addition, we tested whether the observed slight changes in protein fractions lead to an increase or decrease in serum γ, β2, α2, α1 and HSA protein levels expressed in g/L, but there were no changes that reached the significance level.

### 3.5. Serum Protein Profile—The Ratio of Proteins’ Fractions

The ratios of the three most abundant protein fractions (HSA, γ and β) were calculated and presented (Table 4). At time point T1, when dehydration was the dominant process, the HSA/γ ratio obtained was the highest and then decreased to a level below the baseline in placebo. At time points T2 and T3, when proteinuria predominated, increases in the γ/β ratio were obtained (*p* < 0.05), compared to the pre-race value in the placebo group. In contrast, in the aronia group, HSA/γ, HSA/ß and γ/ß ratios remained unchanged 24 h after the race compared to baseline values.

## 4. Discussion

The dehydration and proteinuria caused by PA can alter the levels of serum parameters such as urea, creatinine, uric acid [3] and total proteins [8,9], and could lead to changes in blood protein profiles after PA, such as a half marathon run. The renal function parameters urea, creatinine and uric acid were elevated 15 min and 60 min after the half marathon for all observed parameters. The values of all three parameters observed returned to baseline values in the aronia group, while in the placebo group the urea and uric acid values were still elevated 24 h after the run. Statistically significant differences between the groups were found in urea levels 24 h after the race, with urea being statistically significantly higher in the placebo group than in the aronia group. There was a trend towards an increase in total protein and albumin after the half marathon race, with statistical significance only reached in the aronia group 60 min after the race. When looking at the protein profiles, it was found that there was a statistically significant increase in γ-fractions at the 60 min post-race and 24 h post-race time points in the placebo group. Based on the results, the consumption of polyphenol-rich aronia juice could induce a benign, transient change in renal function during exercise, and reduce exercise-induced protein loss that causes exercise-induced proteinuria. The results obtained suggest that the acute consumption of polyphenols before intense physical activity may influence protein loss during intense physical activity.

PA can lead to acute changes in renal function, a decrease in glomerular filtration rate, and tubular dysfunction [39,40] as well as an accumulation of creatinine, urea, and uric acid in the blood [3,41]. It was found that the concentrations of urea, creatinine and uric acid were significantly higher 15 min and 1 h after the race compared to the baseline values in both groups (placebo and aronia) (Table 1). In clinical practice, the diagnosis of acute kidney injury is only made based on changes in creatinine levels [3]. Since creatinine levels were highest immediately after the race (15 min) and were significantly elevated at 1 h after the race, and returned to baseline levels in both groups 24 h after the race, the changes in these parameters may indicate that transient changes in renal function occurred after the race. Muscle wasting during and after PA could also lead to an increase in serum creatinine concentration. However, according to the literature data, this event occurs later compared to renal injury and should lead to an increase in creatinine levels 24 h after the race [3]. In the placebo group, urea and uric acid levels were significantly higher 24 h after the race (*p* ˂ 0.01), whereas in the aronia group they returned to baseline values. Urea and uric acid levels reflect not only acute kidney injury, but also increased metabolism, dehydration, and hormonal activation during PA [3]. So, besides PA, increased metabolism and increased purine metabolism can also be a source of increased urea and uric acid, respectively [42,43]. According to the obtained results for total proteins and albumins in time points T1 and T2, dehydration could induce changes in these parameters of about 5-9% (obtained percentages for urea and uric acid around 16 and 12%, respectively), but it could be a contributing factor, and probably not the only one. Although there were no differences between the treatment groups, urea levels returned to baseline values only in the aronia group 24 h after the race, suggesting that the consumption of polyphenol-rich juices may have beneficial effects on kidney function. There are several studies that have shown the positive effects of aronia polyphenol extracts on renal function in animal models [20,44]. In addition, two recent studies have reported the beneficial effects of aronia product consumption, including those on renal function parameters, in patients with chronic kidney disease on hemodialysis treatment [45] and in type 2 diabetics [46]. Further studies with a larger number of participants should confirm the potentially protective effect of aronia juice on PA-induced acute kidney injury.

An analysis of the changes in serum protein profiles and serum total protein, HSA and non-albumin protein levels may allow some conclusions to be drawn about the effects of dehydration and the occurrence of proteinuria after a half marathon race. If dehydration is the cause of the increase in total proteins immediately after PA, the protein profiles would be the same as before the race. If proteinuria is the cause of the decrease in total proteins 24 h and 7 days after the first race, then the protein profiles would be different. The obtained trend of transient changes in total proteins and HSA concentrations 15 min and 1 h after the race in both groups (Table 2) is consistent with the data in the literature [4,9], and could be explained by dehydration due to PA. The significant decrease in total serum proteins 24 h after the race in the placebo group could be explained by PA-induced proteinuria [8,41]. Although no statistically significant changes were observed in the placebo group at time T1, the concentrations of total proteins (Table 2) and all protein fractions (Appendix A) were higher compared to baseline values, while the percentage abundances of all five protein fractions remained almost at the corresponding pre-race values (Table 3). At the same time, the ratio of HSA to γ and ß increased slightly (Table 4). Taking all the above into account, slight dehydration appears to have occurred, causing a transient increase in total proteins and HSA 15 min after the race in the placebo group (Table 2). This can be explained by the fact that dehydration is closely related to HSA levels, and is even the only known cause of hyperalbuminemia (increase in the HSA fraction in serum compared to the other fractions) in humans [11,12]. From the protein profiles obtained 1 h after the race in the placebo group and the increase in the γ-fraction (*p* < 0.05) compared to baseline (Table 3), it can be concluded that proteinuria could occur in addition to the dehydration still present (total proteins are above baseline, Table 2). In the aronia group, total proteins were significantly (*p* < 0.05) increased at time points T1 and T2 (Table 2), leading to an increase in the content of all protein fractions expressed in g/L (*p* > 0.05, Appendix A), but without significant changes in their percentages (Table 3). According to the data obtained, dehydration was also present 15 min and even 1 h after the race in the aronia group. The differences between the two groups, which were determined for the period of dehydration, could be a consequence of aronia supplementation and its effect on renal function [20,44,45,46].

Proteinuria could lead to selective protein loss in athletes. Because γ-globulins have the highest molecular weights (MW) (from 150–800 kDa) compared to the other serum proteins, and are the second most abundant serum protein fraction after HSA [37], they cannot easily pass glomerular filtration membranes. As intensive PA could lead to renal dysfunction and proteinuria [41], it is expected that proteins with low MW, such as HSA (66.7 kDa) [45], and others present in β, α2 and α1 can pass through the kidney glomeruli easier compared to those from γ-globulins fraction. In this study, the single bout of PA was accompanied by the significant increase of the percentage of γ-fraction (*p* < 0.05) in time points T2 and T3 in the placebo group, but no significant decrease of the fractions α1, α2 and β was revealed. It seems that PA without aronia supplementation can alter the ratio of the most abundant serum protein fractions, such as HSA and γ. This change is likely due to the differential ability of these proteins to be extracted by the kidney during PA. HSA has a molecular weight (MW) of 66.7 kDa and is more likely to be extracted by the kidneys than the gamma fraction proteins, whose MW is between 150 and 500 kDa. However, baseline protein profiles could indicate what alterations could be induced by a combination of repetitive PA proteinuria and inadequate protein intake. It was found that the recreational runners had reduced percentages of α1 and α2 protein fractions (1.6–2.1%, and 1.6-2.3%, respectively) and elevated percentage of the γ fraction (23.4–25.8%) compared to the available literature data for healthy persons who are not exposed to moderate or intensive PA (α1-globulins 3.6-6.8%, α2-globulins 4.8–9.0%, γ-globulins from 10.7–22.5%, HSA 52.7–67.1%, and β-globulins 8.2–13.5%) [47]. The abundance of HSA and β fractions was within the reference range for healthy individuals. The participants in this study are active members of the Belgrade Urban Running Team, who run approximately 40 km per week during 4 practices [27]. So, it can be concluded that the observed changes of total serum proteins and serum protein profiles could be a consequence of repetitive long-term PA-induced proteinuria due to the loss of low MW proteins. It seems that the proteins present in α1 and α2 fractions, and partially in HSA, mainly contribute to the decrease of total serum proteins found in these subjects before the first race and after the first race (Figure 2). According to these results, as well as the fact that the total proteins were even significantly lowered before the second race (Figure 2), it can be concluded that low MW proteins contributed to the total loss of proteins. Because the abundance of α1 fraction non-statistically decreases after a single bout of PA, this fraction could be the most extensively lost by proteinuria. Alpha-1 antitrypsin (MW 52 kDa), thyroxine-binding globulin (MW 54 kDa) and transcortin (MW 52 kDa) are the main components of the α1 fraction. Alpha-1 antitrypsin is the most abundant of them, with a mean concentration in serum of 1.48 ± 0.2 g/L, and its deficiency can result in liver failure and chronic lung diseases [48]. At the same time, a proteomic analysis revealed that there are between 50 and 80 different proteins in the serum, whose molecular weights are between 9.33 kDa and 65kDa [49], and they are parts of three minor protein fractions, which could contribute to total protein decrease after 24h in the placebo group and 7 days after the first race. Several recent proteomic studies in athletes have revealed that concentrations of numerous individual proteins can be affected by PA (1, 2, 3). According to the literature data, there are several PA-induced mechanisms [41] which lead to the increase of renal membrane permeability and transient proteinuria in athletes. As proteinuria induces the selective loss of low MW proteins with unique biological roles, it is imperative to develop nutrition approaches in the management of PA-induced kidney functional changes. In the aronia juice group, the changes in the protein’s profiles was less evident, as there was no change of γ fraction abundance in time point T2 and T3 like in the placebo group (Table 3). In addition, the difference in mean values between T0 and T3 for the HSA fraction is greater in the placebo group (1.4 g/L) than in the aronia group (0.8 g/L) (*p* > 0.05). Finally, the significant difference between the two groups is the observed effect of aronia juice consumption on the total protein content 24 h after the race, as the decrease of total proteins was detected only in the placebo group (*p* < 0.05) (Table 2). So, the obtained results for aronia juice treatment indicates that the acute consumption of aronia juice by recreational runners before the simulated half-marathon race could exert beneficial effects on the level of proteinuria. So far, Macena and el. (2022)’s meta-analysis has reported the modest effect of the intake of dietary polyphenols on proteinuria in diabetic nephropathy [50]. Furthermore, in a recent review article, the authors focused on dietary polyphenols’ effects on mitochondrial damage, that had been found to be associated with kidney damage in different experimental models [51]. Finally, flavonoids, a class of polyphenol molecules, have shown potential to prevent dysfunction and improve the function of kidneys by influencing oxidative stress and inflammation pathways (Cao, 2022) [52]. Although there is no literature data that aronia consumption could show an effect in athletes on PA-induced proteinuria (review article effects of aronia), the obtained findings are in line with previous studies exploring the effects of polyphenols on kidney function [50,51,52].

This study showed that a period of seven days between two half-marathon races did not allow for the complete recovery of total proteins on the level before the first race (Figure 2). Furthermore, it seems that besides confirmed proteinuria, inadequate dietary protein consumption by the participants involved in this study probably contributes to a significant decrease of total proteins in the period between two races. Our previous study revealed that 8/10 subjects had inadequate dietary protein intake compared to the values recommended by the International Society of Sports Nutrition for exercising individuals [27]. Before the first race, 3/10 participants had total protein content below the normal reference range (64–83 g/L), while in 7/10 participants, these values were only slightly above the lower limit. Although the obtained mean value of total proteins was low (65.5 ± 3.5 g/L) before the first race, the mean value of HSA (39.8 ± 3.6 g/L) was in the normal reference range (35–55 g/L). Because HSA is the most abundant protein in the serum (60% of total serum proteins), only one third of the total HSA content is found in the intravascular department and has a half-life of 20 days [53,54,55]; its loss by proteinuria, caused by repetitive PA, is less evident but still present (mean value 39.8 g/L was closer to the low reference value 35 g/L). According to these results, it can be concluded that repetitive PA-induced proteinuria and inadequate protein intake by recreational runners could lead to the decrease of total serum proteins, including both HSA and non-albumin proteins with low MW, such as alpha-1 antitrypsin, thyroxine-binding globulin and transcortin. 

### Limitations

Our study has some limitations that could have an impact on the interpretation of results. We were not able to obtain the data on the participants’ previous polyphenol intake due to the limitations of the Serbian Food Composition Database, which we have used to assess the participants’ dietary habits, as described previously [27]. Still, the participants were advised to refrain from consuming polyphenol-rich foods before the intervention. Also, in our dietary assessments, we considered the consumption of major antioxidants, which was found to be inadequate. Thus, as we suggested previously [27], there were not any antioxidants consumed in excess that could influence the interpretation of our findings. Another limitation of our research (due to financial constraints) is not performing VO2 max measurements, which is an important determinant of running performance. Finally, another important limitation is the small sample size of our study group. However, we counteracted these limitations by recruiting healthy male volunteers from the same team and following the same training regime for enough time before the intervention. Also, we applied the cross-over design, meaning each subject serves as his own control, allowing us to minimize the possible effects of confounding covariates on our findings.

## 5. Conclusions

Our pilot study showed that acute aronia juice consumption before race could favorably influence the loss of serum proteins and affect PA-induced proteinuria. So, this, together with an adequate protein intake, could be a healthcare measure for the prevention of the benign, transiting changes of PA-induced kidney function alterations in recreational runners, and, as repeated, regular PA could lead to loss of low-molecular protein.

## Figures and Tables

**Figure 1 healthcare-12-01276-f001:**
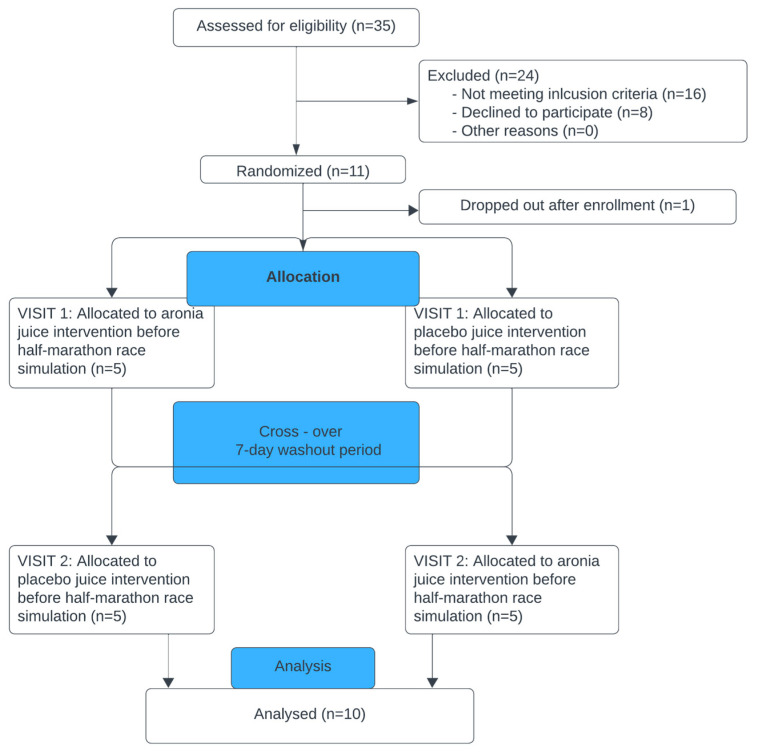
Summary of research design.

**Figure 2 healthcare-12-01276-f002:**
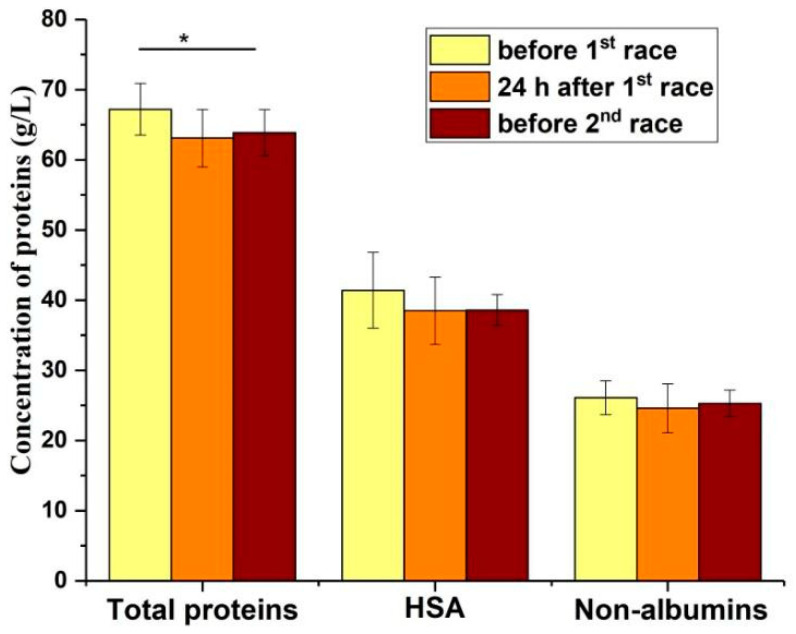
Total proteins, HSA, and non-albumin proteins (g/L) in serum before, 24 h and 7 days after the first simulated half-marathon race. * *p* ˂ 0.05 compared to value before 1st race.

**Table 1 healthcare-12-01276-t001:** Values of urea, creatinine, and uric acid before (T0) and 15 min (T1), 1 h (T2) and 24 h (T3) after race.

Group	Time	Urea (mM)	Creatinine (µM)	Uric Acid (mM)
Placebo	T0	4.4 ± 1.3	81 ± 12	313 ± 50
	T1	5.1 ± 1.3 **	94 ± 14 **	343 ± 50 ***
	T2	5.1 ± 1.2 *	89 ± 13 **	353 ± 54 ***
	T3	5.1 ± 1.2 **#	83 ± 12	341 ± 42 **
Aronia juice	T0	4.9 ± 1.7	83 ± 13	335 ± 49
	T1	5.4 ± 1.7 **	91 ± 17 **	357 ± 56 ** ##
	T2	5.4 ± 1.7 **	89 ± 16 *	353 ± 60 *
	T3	5.0 ± 1.3	85 ± 12	359 ± 69

* *p* ˂ 0.05, ** *p* ˂ 0.01, *** *p* ˂ 0.001 (paired *t*-test) compared to corresponding baseline values (T0). # *p* ˂ 0.05. ## *p* ˂ 0.01 compared to the treatment group, normalized values.

**Table 2 healthcare-12-01276-t002:** Total proteins, HSA, and non-albumin proteins content (g/L) in serum.

Group	Time	Total Proteins (g/L)	HSA (g/L)	Non-Albumins (g/L)
Placebo	T0	67.2 ± 3.7	41.1 ± 5.1	26.1 ± 2.4
	T1	70.9 ± 4.5	44.0 ± 6.2	26.9 ± 3.3
	T2	69.5 ± 3.1	41.4 ± 4.1	28.1 ± 3.6
	T3	63.1 ± 4.1 *	38.5 ± 4.8	24.6 ± 3.5
Aronia juice	T0	63.9 ± 3.3	38.6 ± 2.2	25.3 ± 1.9
	T1	67.5 ± 4.4 *	40.5 ± 4.1	27.1 ± 1.7
	T2	69.2 ± 3.1 *	42.1 ± 4.1 *	27.2 ± 3.2
	T3	65.1 ± 2.5	38.9 ± 3.5	26.2 ± 3.9

* *p* < 0.05 (paired *t*-test) compared to corresponding baseline values (T0).

**Table 3 healthcare-12-01276-t003:** The abundance (%) of albumin (HSA) and non-albumin fractions (gamma, beta, alpha_1_, and alpha_2_) in serum.

Group	Time	Abundance of Serum Protein Fractions (%)				
		Gamma	Beta	Alpha_2_	Alpha_1_	Albumin
Placebo	T0	22.8 ± 4.2	11.3 ± 1.8	2.0 ± 0.9	1.9 ± 1.1	62.2 ± 5.8
	T1	22.0 ± 5.4	10.9 ± 1.3	2.2 ± 0.9	1.9 ± 0.8	63.0 ± 5.9
	T2	24.7 ± 4.0 *	10.5 ± 1.4	2.3 ± 1.3	2.0 ± 1.0	60.6 ± 5.1
	T3	25.0 ± 4.0 *	11.0 ± 1.8	2.0 ± 0.6	1.2 ± 0.7	60.8 ± 5.1
Aronia juice	T0	25.7 ± 3.1 +	11.1 ± 2.0	1.4 ± 1.0	1.3 ± 1.2+	60.6 ± 1.4
	T1	25.6 ± 2.5	11.3 ± 1.7	1.4 ± 0.6	1.8 ± 0.7	59.9 ± 2.8
	T2	26.0 ± 3.0	10.8 ± 1.8	1.2 ± 1.0	1.2 ± 0.6	60.8 ± 4.6
	T3	26.3 ± 4.7	11.0 ± 1.0	1.7 ± 1.0	1.2 ± 0.7	59.8 ± 5.4

* *p* < 0.05 compared to the baseline value; + *p* < 0.05 compared to the baseline in placebo group.

**Table 4 healthcare-12-01276-t004:** The ratios of HSA/gamma, HSA/beta and gamma/beta fractions.

Group	Time	HSA/Gamma	HSA/Beta	Gamma/Beta
Placebo	T0	2.8 ± 0.8	5.7 ± 1.2	2.0 ± 0.4
	T1	3.1 ± 1.3	5.9 ± 1.0	2.0 ± 0.5
	T2	2.55 ± 0.7	5.9 ± 4.1	2.3 ± 0.3 *
	T3	2.5 ± 0.7	5.7 ± 1.3	2.3 ± 0.5 *
Aronia juice	T0	2.4 ± 0.3	5.6 ± 0.8	2.4 ± 0.6
	T1	2.4 ± 0.3	5.4 ± 0.9	2.3 ± 0.4
	T2	2.4 ± 0.5	5.8 ± 1.3	2.5 ± 0.5
	T3	2.4 ± 0.7	5.5 ± 0.8	2.4 ± 0.5

* *p* < 0.05 compared to baseline value.

## Data Availability

The raw data supporting the conclusions of this article will be made available by the authors on request.

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
