# Peer review of "The Potential Benefits of Acute Aronia Juice Supplementation on Physical Activity Induced Alterations of the Serum Protein Profiles in Recreational Runners: A Pilot Studyâ€"

_healthcare, 2024, doi:10.3390/healthcare12131276_

Round 1
Reviewer 1 Report (Previous Reviewer 1)
Comments and Suggestions for Authors
Author have incorporated relevant information. However in the response to comments, author must mention the line number for smooth review of article.
Minor revisions
1)Author have incorporated relevant information. However in the response to comments, author must mention the line number for smooth review of article
2) Author must check grammatical mistake throughout the manuscript

English language must be edited moderately
Author Response
Respond to the Reviewers
Dear Reviewers,
We sincerely thank you for your thorough and valuable review. We greatly appreciate your detailed comments, which have provided a clear path to improve our manuscript. We are grateful for the opportunity to refine our work considering your suggestions. We are confident that incorporating these changes will significantly enhance the quality of the article. The new changes made in the text (except for English editing) were marked green.
Your expertise and thoughtful guidance are immensely valued, and we are excited to enhance our manuscript in line with your suggestions, hoping that it will be a valuable contribution to the journal published by MDPI.
Sincerely,
Authors
Reviewer 1
Comments and Suggestions for Authors
Author have incorporated relevant information. However in the response to comments, author must mention the line number for smooth review of article.
Minor revisions
Comment 1. Author have incorporated relevant information. However in the response to comments, author must mention the line number for smooth review of article
Response: We apologize for not marking induced changes appropriately. We hope that this modified version of review will allow smooth review.
Article: The Potential Benefits of Acute Aronia Juice Supplementation 2 on Physical Activity Induced Alterations of the Serum Protein 3 Profiles in Recreational Runners: A Pilot Study
Comments
Introduction:
1) The source of Aronia juice, its growing condition, availability in the countries, trade name, varieties of Aronia available, variety Aronia juice available commercially, a component of Aronia, etc. information must be mentioned along with references.
Your suggestion for a more detailed description of Aronia juice origin was considered and appropriate changes were incorporated into the manuscript in the introduction section and material and methods.
lines 79-87 Aronia (Aronia melanocarpa (Michx.) Elliot), is a shrub cultivated worldwide. It gained the greatest popularity in the northern, eastern (including Serbia), and central countries of Europe. Aronia grows in a wide variety of conditions, as it is highly adaptable giving a high yield. It can grow in mild climatic conditions, but also at extremely low temperatures [16]. The composition of aronia fruits depends on various factors, such are maturity, climatic, and environmental conditions. Aronia fruits are rich sources of phenolic compounds, including anthocyanins, phenolic acids, proanthocyanidins, and other subclasses. Polyphenols are compounds that govern the high bioactivity of aronia berries [17].
lines 125-129 The commercially available aronia (black chokeberry) Nero cultivar juice was donated from Rheapharm d.o.o., Belgrade, Serbia. Briefly, the juice was made by mechanical processing, including homogenization and pressing, from ripe, carefully selected fruits of Aronia melanocarpa, harvested in August/September from a plantation field in the mountain Suvobor (750 m a.s.l.), Serbia.
references 16, 17
- Gerasimov, M.A.; Perova, I.B.; Eller, K.I.; Akimov, M.Y.; Sukhanova, A.M.; Rodionova, G.M.; Ramenskaya GV. Investigation of polyphenolic compounds in different varieties of black chokeberry Aronia melanocarpa. Molecules. 2023, 28, 4101. doi: 10.3390/molecules28104101
- Sidor, A.; Gramza-Michałowska, A. Black Chokeberry Aronia melanocarpa L.-A qualitative composition, phenolic profile and antioxidant potential. Molecules. 2019, 24, 3710. doi: 10.3390/molecules24203710.
2) Elaborate “Proteinuria is a benign phenomenon” , hyperalbuminemia, along with references
According to the suggestion, the text was changed.
lines 72-74 In PA, proteinuria is considered as a benign phenomenon, because it is not related to any structural changes in the kidney [3]. This transient functional change of the kidney is caused by different mechanisms [14].
3) What are the different markers of kidney injury? It must be mentioned along with references.
Some markers of kidney injury were added in the text.
lines 65-68 Different parameters are recommended as the markers of kidney injury such as serum level of creatinine, urea and uric acid or albuminuria according to Kidney Disease Improving Global Outcomes criteria [13].
reference 13. Ad-Hoc Working Group of ERBP; Fliser, D.; Laville, M.; Covic, A.; Fouque, D.; Vanholder, E.; Juillard, L.; Van Biesen, W. A European Renal Best Practice (ERBP) position statement on the Kidney Disease Improving Global Outcomes (KDIGO) clinical practice guidelines on acute kidney injury: Part 1: Definitions, conservative management and contrast-induced nephropathy. Nephrol. Dial. Transplant. 2012, 27, 4263–4272. doi: 10.1093/ndt/gfs375.
Methods
1) How it was decided to provide 200 mL of aronia juice to all the participants? Similarly, how it was decided whether before/after breakfast or before/after exercise these juice will be provided to all the participants. Also whether fresh juice or commercially available juice is needed to be provided to the participants. Information must be incorporated along with references.
All suggestions were accepted and the text below and relevant references were added to the manuscript.
lines 132-142 The juice from the same batch was provided to all the participants. The juice/placebo was given shortly after breakfast because of its strong astringent taste and to prevent it from causing an upset stomach. Based on previous studies that investigated the acute effects of polyphenol-rich sources on exercise-induced oxidative stress, we decided to give aronia/placebo juice to the participants before exercise [29, 30, 31]. In addition, the timing of aronia consumption was chosen, so that peak anthocyanin and total polyphenols concentration would occur in blood during the exercise [32]. The dose of the juice approximately matched the polyphenol content in other polyphenol-rich fruit juices that were previously investigated in recreational and elite runners [33, 34].
references
- Decroix, L.; Tonoli, C.; Soares, D.D.; Descat, A.; Drittij-Reijnders, M.J.; Weseler, A.R.; Bast, A.; Stahl, W.; Heyman, E.; Meeusen, R. Acute cocoa Flavanols intake has minimal effects on exercise-induced oxidative stress and nitric oxide production in healthy cyclists: a randomized controlled trial. J. Int. Soc. Sports Nutr. 2017, 14, 28. doi: 10.1186/s12970-017-0186-7
- Davison, G.; Callister, R.; Williamson, G.; Cooper, K.A; Gleeson M. The effect of acute pre-exercise dark chocolate consumption on plasma antioxidant status, oxidative stress and immunoendocrine responses to prolonged exercise. Eur. J. Nutr. 2012, 51, 69-79. doi: 10.1007/s00394-011-0193-4.
- Morillas-Ruiz, J.; Zafrilla, P.; Almar, M.; Cuevas, M.J.; López, F.J.; Abellán, P.; Villegas, J.A.; González-Gallego, J. The effects of an antioxidant-supplemented beverage on exercise-induced oxidative stress: results from a placebo-controlled double-blind study in cyclists. Eur. J. Appl. Physiol. 2005, 95, 543-9. doi: 10.1007/s00421-005-0017-4.
- Xie, L.; Lee, S.G.; Vance, T.M.; Wang, Y.; Kim, B.; Lee, J.Y.; Chun, O.K.; Bolling, B.W. Bioavailability of anthocyanins and colonic polyphenol metabolites following consumption of aronia berry extract. Food Chem. 2016, 211, 860-8. doi: 10.1016/j.foodchem.2016.05.122.
- Howatson, G.; McHugh, M.P.; Hill, J.A.; Brouner, J.; Jewell, A.P.; van Someren, K.A.; Shave, R.E.; Howatson, S.A. Influence of tart cherry juice on indices of recovery following marathon running. Scand. J. Med. Sci. Sports. 2010, 20, 843-52. doi: 10.1111/j.1600-0838.2009.01005.x.
- Dimitriou, L.; Hill, J.A.; Jehnali, A.; Dunbar, J.; Brouner, J.; McHugh, M.P.; Howatson, G. Influence of a montmorency cherry juice blend on indices of exercise-induced stress and upper respiratory tract symptoms following marathon running--a pilot investigation. J. Int. Soc. Sports Nutr. 2015, 12, 22. doi: 10.1186/s12970-015-0085-8.
2) How blood samples were collected and how much quantity of samples per participants were collected.
Information was provided in the 2.2 section.
lines 150-153 After an overnight fast before breakfast, a venous blood sample was collected before (T0), and 15 min (T1), 1 h (T2), and 24 h (T3) after each race into serum sample tubes. One tube was collected per participant at each time-point, a total of four tubes per race. Prior to each blood sampling, participants rested for at least 10 min.
3) How placebo juice was made without polyphenols. Brief information must be made.
Information on placebo juice was added in the M&M section.
lines 142-146 The placebo juice was originally developed as a control beverage to be used in intervention studies with aronia juice. More precisely, three formulations were prepared by mixing various nutrients, matching the nutrient composition of aronia juice, in water, with the addition of artificial colors and flavors. As previously published, the similarity of these formulations to aronia juice was assessed by six food panelists so that formulation was chosen
4) Whether blood was drawn from the same participants at different time points, information must be included.
Information was included in 2.2 section.
lines 150-153 After an overnight fast before breakfast, a venous blood sample was collected before (T0), and 15 min (T1), 1 h (T2), and 24 h (T3) after each race into serum sample tubes. One tube was collected per participant at each time-point, a total of four tubes per race. Prior to each blood sampling, participants rested for at least 10 min.
5) In this study assays were conducted in duplicate. However, it is extremely essential to conduct each experiment in triplicate.
All assays were conducted in triplicate as it was stated in section 2.6 Statistical analysis except determination of protein fractions by electrophoresis that was performed in duplicate. Determination of serum protein profiles is a standard biochemical analysis that is routinely performed in clinical laboratories in monoplicate. So, we considered that it would be acceptable to perform the assay in duplicate. Furthermore, it was justified by obtaining a low CV value (< 5%) for the analysis.
6) While determination of serum protein profiles, by which method protein content in the sample was estimated, why native gel and why 9 % PAA gel was used. What is PAA. Incorporate information along with references.
A more detailed description considering the electrophoretic analysis of protein profiles was added in Section 2.4. in line with your suggestions.
lines 166-173 Native polyacrylamide gel electrophoresis (PAGE) was performed according to the manufacturer’s recommendations using a Hoefer SE 260 electrophoretic unit (San Francisco, CA, USA) for the separation of five main serum protein fractions (gamma, beta, alpha2, alpha1 and HSA) [37]. Before electrophoresis, the concentration of proteins in the serum was determined by the biuret method [35]. The experiment was performed in duplicate. In the first experiment 5 µg of serum proteins was applied per line, while in the second 7 µg of proteins was applied on 9 % polyacrylamide gel for the better separation of single serum protein fractions.
New reference number 37 was also added.
- O'Connell, T.X.; Horita, T.J.; Kasravi, B. Understanding and interpreting serum protein electrophoresis. Am. Fam. Physician. 2005, 71, 105-12. PMID: 15663032.
7) Method for determination of urea, creatinine and uric acid must be incorporated along with references.
Appropriate information on the methods for the determination of urea, creatinine, and uric acid are added in section 2.5.
lines 179-190 Briefly, creatinine and uric acid were determined by enzymatic colorimetric methods. Creatinine was converted into glycine, formaldehyde and hydrogen peroxide using creatininase, creatinase, and sarcosine oxidase, while uric acid was converted into allantoin and hydrogen peroxide using uricase. In the next step, the liberated hydrogen peroxide was converted into quinone-imine chromogen by action of peroxidase, whose color intensity was directly proportional to the creatinine or uric acid concentration (Roche Diagnostics, Basel, Switzerland). Urea was measured by the kinetic enzymatic assay in which urea was hydrolyzed by urease into carbonate and ammonium. In the next step, these products were converted into L-glutamate in the presence of glutamate dehydrogenase and coenzyme NADH. The rate of decrease in the concentration of NADH was directly proportional to the concentration of urea in the sample [38].
- Sampson, E.J.; Baird, M.A.; Burtis, C.A.; Smith, E.M.; Witte, D.L.; Bayse, D.D. A coupled-enzyme equilibrium method for measuring urea in serum: optimization and evaluation of the AACC study group on urea candidate reference method. Clin. Chem. 1980, 26, 816-26. PMID: 7379302.
Results:
1) The importance of each experiment conducted, its analysis etc must be described in the results section.
Thank you for your valuable suggestion, the corrections are introduced in the text and marked as red letters.
2) The values in Table 3 are not properly formatted.
We corrected that and appropriately formatted other Tables, too.
3) Relevant changes in protein fractions in all groups must be mentioned.
The suggestion is accepted, thank You.
lines 271-287 According to the results presented in Table 3, the abundances of γ and β obtained at time point T1 were almost unchanged compared to the baseline values, while the trends of changes in the abundances of α2, α1 and HSA were different in both groups. At time points T2 and T3, an increase in the percentage of the γ fraction was observed in both groups compared to baseline values, but this was only statistically significant (pË‚0.05) in the placebo group. Looking at the non-statistically significant changes in T2 and T3, α1 fraction abundance decreases in the placebo group in T3. In addition, the difference in mean values between T0 and T3 for the HSA fraction is greater in the placebo group (1.4 g/L) than in the chokeberry group (0.8 g/L) (p>0.05).
According to these results, HSA and α1 could contribute to a significant decrease in total serum proteins 24 h after the race in the placebo group (Table 2). In addition, the decrease in these protein fractions is probably jointly responsible for the decrease in total proteins 24 h and 7 days after the first race (Figure 2). The decrease in total proteins is likely due to proteinuria, as both total protein levels and protein profiles were altered.
In addition, we tested whether the observed slight changes in protein fractions lead to an increase or decrease in serum γ, β2, α2, α1 and HSA protein levels expressed in g/L (results not shown), but there were no changes that reached the significance level.
4) The sample size should be large enough to conclude the results appropriately.
We have agreed with the reviewer that 10 participants included in the study was not enough for strong statistical power and a higher number of participants was needed for that. In this pilot study, we tried to investigate how long-term moderate PA (runners who practice running 40 km per week) together with the simulated half-marathon races influence serum protein profiles. The number of participants from the Belgrade running club who fulfilled this requirement and wanted to participate in the study was limited to 10. We know that this pilot study does not have strong statistical power, but can offer the trend of changes in serum protein profiles. This study can be important because it offers a foundation for further studies in which a higher number of recreational participants will be included. The results of this study may be important, because they point out the possible negative effect of moderate or intensive long-term PA on the health of recreational runners, especially on those who are not under the control of professional nutritionists, trainers, and doctors or may not consume recommended quantities of nutrients or supplements which help them to recover from the negative effects of PA.
Discussion:
1) Discussion not written appropriately
2) Results needed to be analyzed, interpreted sufficiently and discussed for appropriate
conclusion.
3) All physiological parameters must be discussed.
All suggestions were accepted and the discussion section was carefully restructured and the missing parts were added. We hope that induced changes significantly improved the discussion section quality.
4) It is written in line 274) “….proteins with LMW such as HSA …”, here what is Mol wt of HSA? Please Mention
The molecular weight of HSA is reported to be 66700 Da and it was added in text along with the relevant reference.
- Lee, P.; Wu, X. Review: modifications of human serum albumin and their binding effect. Curr. Pharm. Des. 2015, 21, 1862-5. doi: 10.2174/1381612821666150302115025.
5) Why specifically α1, α2, γ-globulins levels of participants measured in this study? It must have been discussed along with reference literature.
The content of protein fractions α1, α2, ß, γ-globulins and HSA was determined as they represent the main protein fractions in serum after separation by electrophoresis. The relevant reference (number 37, O'Connell et al, 2005) was added in the text of the revised version of the manuscript (lines 260-261 as well as in method and material section 2.4.).
lines 260-261 It is known that non-albumin proteins can be separated into four different fractions by native electrophoresis: alpha1 (α1), alpha2 (α2), beta (ß) and gamma (γ) [37].
- O'Connell, T.X.; Horita, T.J.; Kasravi, B. Understanding and interpreting serum protein electrophoresis. Am. Fam. Physician. 2005, 71, 105-12. PMID: 15663032.
6) It is written that (line 316) “…. proteinuria and in-adequate protein intake could show a more pronounced effect on some non-albumin proteins than on HAS”, here please incorporate few non-albumin proteins' name and their references should have been mentioned.
The names of several low molecular proteins were added.
7) Reference for the line 337 “HSA is the major transport … of great importance” must be mentioned
The discussion section was carefully rewritten.
Conclusion
Conclusion of study has not been drawn appropriately. Several gaps are there. For ex what is the minimum time required for recovery and symptoms of those who are continuously doing PA? what does Acute aronia juice consumption means, Case study etc. must also discussed.
The conclusion section was carefully rewritten.
Others
Full form of all abbreviations (PAA, CBB, LDH, CK,LMW) must be mentioned when appear 1st time in the manuscript.
The abbreviations were checked.
Comment 2) Author must check grammatical mistake throughout the manuscript
Response: The English language was checked by a native English speaker.
Reviewer 2
Comments and Suggestions for Authors
- L24-25 – Abstract: “The significant changes in urea, creatinine and uric acid levels were noticed at 24 selected time points.” In what groups? Need to specify.
Response: Thank You for a suggestion we have specified that changes were noticed in both groups (marked green).
- Overall, the paper needs to be proofread to improve English grammar and readability. For example, L34 – “It is defined as a “PA…” What is It? Comes across vague.
Response: The English editing was done by a native English speaker.
- L100-102 – The goal of the study stated is confusing. If the aronia juice is the treatment, then “effect of consuming aronia juice Vs placebo in people performing PA” should be the focus of the study. Currently it states, “ The main goal of this pilot single-blind crossover placebo-controlled study was to examine the effect of repetitive PA with or without acute aronia juice supplementation on serum protein profiles of recreational runners running 40 km each week in selected time points before and after simulated half-marathon races”. The term repetitive PA is confusing. How was repetitive PA measured? Is that a variable? The sentence is too long. As an example, goals could be written - “The primary goal of this pilot single-blind crossover placebo-controlled study was to examine the effect of aronia juice supplementation versus placebo on serum protein profiles in male recreational runners at selected time points before and after simulated half-marathon races” This can be followed by listing other variables or secondary variables that was measured. Avoid terms like “for the more deep insight”. “More deep” are repetitive adjectives, hence redundant. Simply state what additional parameters were measured.
Response: Thank You for a valuable suggestion the lines 100-102 was carefully rewritten and the sentence line 106-107 modified (marked green).
- Throughout the document aronia group and chokeberry group are used interchangeably. Please be consistent and identify the treatment group with just one name.
Response: We apologize for the inconsistency. The manuscript was carefully checked and term aronia is used.
- Line 254-262 – “To confirm that two intensive PA….” is very confusing. This is the first time that the study involves 2 marathons is discussed. Please briefly mention the timeline along with number of PA in the methods section.
Response: The following sentence (marked green) was added in the section 2.1, lines 123-125 to support the text lines 254-262 :
The wash-out period between two simulated half-marathon races that each participant ran was 7 days (Figure 1).
- Discussion – Is reworded well, however, some more introspect into results can strengthen the discussion section. For example, L296-299 can be moved to discussion section, and this will be a value add. Review the results and move such descriptive and conclusive statements to the discussion section
Response: Thank You for the suggestion, we replaced several sentences from results to discussion section (marked green).
Comments on the Quality of English Language
Needs grammar review. Some sentences are too long and some start vaguely.
Response: The English language was checked by a native English speaker.

Reviewer 2 Report (Previous Reviewer 2)
Comments and Suggestions for Authors
1. L24-25 – Abstract: “The significant changes in urea, creatinine and uric acid levels were noticed at 24 selected time points.” In what groups? Need to specify.
2. Overall, the paper needs to be proofread to improve English grammar and readability. For example, L34 – “It is defined as a “PA…” What is It? Comes across vague.
3. L100-102 – The goal of the study stated is confusing. If the aronia juice is the treatment, then “effect of consuming aronia juice Vs placebo in people performing PA” should be the focus of the study. Currently it states, “ The main goal of this pilot single-blind crossover placebo-controlled study was to examine the effect of repetitive PA with or without acute aronia juice supplementation on serum protein profiles of recreational runners running 40 km each week in selected time points before and after simulated half-marathon races”. The term repetitive PA is confusing. How was repetitive PA measured? Is that a variable? The sentence is too long. As an example, goals could be written - “The primary goal of this pilot single-blind crossover placebo-controlled study was to examine the effect of aronia juice supplementation versus placebo on serum protein profiles in male recreational runners at selected time points before and after simulated half-marathon races” This can be followed by listing other variables or secondary variables that was measured. Avoid terms like “for the more deep insight”. “More deep” are repetitive adjectives, hence redundant. Simply state what additional parameters were measured.
4. Throughout the document aronia group and chokeberry group are used interchangeably. Please be consistent and identify the treatment group with just one name.
5. Line 244-252 – “To confirm that two intensive PA….” is very confusing. This is the first time that the study involves 2 marathons is discussed. Please briefly mention the timeline along with number of PA in the methods section.
6. Discussion – Is reworded well, however, some more introspect into results can strengthen the discussion section. For example, L296-299 can be moved to discussion section, and this will be a value add. Review the results and move such descriptive and conclusive statements to the discussion section.
Comments on the Quality of English LanguageNeeds grammar review. Some sentences are too long and some start vaguely.
Author Response
Respond to the Reviewers
Dear Reviewers,
We sincerely thank you for your thorough and valuable review. We greatly appreciate your detailed comments, which have provided a clear path to improve our manuscript. We are grateful for the opportunity to refine our work considering your suggestions. We are confident that incorporating these changes will significantly enhance the quality of the article. The new changes made in the text (except for English editing) were marked green.
Your expertise and thoughtful guidance are immensely valued, and we are excited to enhance our manuscript in line with your suggestions, hoping that it will be a valuable contribution to the journal published by MDPI.
Sincerely,
Authors
Reviewer 1
Comments and Suggestions for Authors
Author have incorporated relevant information. However in the response to comments, author must mention the line number for smooth review of article.
Minor revisions
Comment 1. Author have incorporated relevant information. However in the response to comments, author must mention the line number for smooth review of article
Response: We apologize for not marking induced changes appropriately. We hope that this modified version of review will allow smooth review.
Article: The Potential Benefits of Acute Aronia Juice Supplementation 2 on Physical Activity Induced Alterations of the Serum Protein 3 Profiles in Recreational Runners: A Pilot Study
Comments
Introduction:
1) The source of Aronia juice, its growing condition, availability in the countries, trade name, varieties of Aronia available, variety Aronia juice available commercially, a component of Aronia, etc. information must be mentioned along with references.
Your suggestion for a more detailed description of Aronia juice origin was considered and appropriate changes were incorporated into the manuscript in the introduction section and material and methods.
lines 79-87 Aronia (Aronia melanocarpa (Michx.) Elliot), is a shrub cultivated worldwide. It gained the greatest popularity in the northern, eastern (including Serbia), and central countries of Europe. Aronia grows in a wide variety of conditions, as it is highly adaptable giving a high yield. It can grow in mild climatic conditions, but also at extremely low temperatures [16]. The composition of aronia fruits depends on various factors, such are maturity, climatic, and environmental conditions. Aronia fruits are rich sources of phenolic compounds, including anthocyanins, phenolic acids, proanthocyanidins, and other subclasses. Polyphenols are compounds that govern the high bioactivity of aronia berries [17].
lines 125-129 The commercially available aronia (black chokeberry) Nero cultivar juice was donated from Rheapharm d.o.o., Belgrade, Serbia. Briefly, the juice was made by mechanical processing, including homogenization and pressing, from ripe, carefully selected fruits of Aronia melanocarpa, harvested in August/September from a plantation field in the mountain Suvobor (750 m a.s.l.), Serbia.
references 16, 17
- Gerasimov, M.A.; Perova, I.B.; Eller, K.I.; Akimov, M.Y.; Sukhanova, A.M.; Rodionova, G.M.; Ramenskaya GV. Investigation of polyphenolic compounds in different varieties of black chokeberry Aronia melanocarpa. Molecules. 2023, 28, 4101. doi: 10.3390/molecules28104101
- Sidor, A.; Gramza-Michałowska, A. Black Chokeberry Aronia melanocarpa L.-A qualitative composition, phenolic profile and antioxidant potential. Molecules. 2019, 24, 3710. doi: 10.3390/molecules24203710.
2) Elaborate “Proteinuria is a benign phenomenon” , hyperalbuminemia, along with references
According to the suggestion, the text was changed.
lines 72-74 In PA, proteinuria is considered as a benign phenomenon, because it is not related to any structural changes in the kidney [3]. This transient functional change of the kidney is caused by different mechanisms [14].
3) What are the different markers of kidney injury? It must be mentioned along with references.
Some markers of kidney injury were added in the text.
lines 65-68 Different parameters are recommended as the markers of kidney injury such as serum level of creatinine, urea and uric acid or albuminuria according to Kidney Disease Improving Global Outcomes criteria [13].
reference 13. Ad-Hoc Working Group of ERBP; Fliser, D.; Laville, M.; Covic, A.; Fouque, D.; Vanholder, E.; Juillard, L.; Van Biesen, W. A European Renal Best Practice (ERBP) position statement on the Kidney Disease Improving Global Outcomes (KDIGO) clinical practice guidelines on acute kidney injury: Part 1: Definitions, conservative management and contrast-induced nephropathy. Nephrol. Dial. Transplant. 2012, 27, 4263–4272. doi: 10.1093/ndt/gfs375.
Methods
1) How it was decided to provide 200 mL of aronia juice to all the participants? Similarly, how it was decided whether before/after breakfast or before/after exercise these juice will be provided to all the participants. Also whether fresh juice or commercially available juice is needed to be provided to the participants. Information must be incorporated along with references.
All suggestions were accepted and the text below and relevant references were added to the manuscript.
lines 132-142 The juice from the same batch was provided to all the participants. The juice/placebo was given shortly after breakfast because of its strong astringent taste and to prevent it from causing an upset stomach. Based on previous studies that investigated the acute effects of polyphenol-rich sources on exercise-induced oxidative stress, we decided to give aronia/placebo juice to the participants before exercise [29, 30, 31]. In addition, the timing of aronia consumption was chosen, so that peak anthocyanin and total polyphenols concentration would occur in blood during the exercise [32]. The dose of the juice approximately matched the polyphenol content in other polyphenol-rich fruit juices that were previously investigated in recreational and elite runners [33, 34].
references
- Decroix, L.; Tonoli, C.; Soares, D.D.; Descat, A.; Drittij-Reijnders, M.J.; Weseler, A.R.; Bast, A.; Stahl, W.; Heyman, E.; Meeusen, R. Acute cocoa Flavanols intake has minimal effects on exercise-induced oxidative stress and nitric oxide production in healthy cyclists: a randomized controlled trial. J. Int. Soc. Sports Nutr. 2017, 14, 28. doi: 10.1186/s12970-017-0186-7
- Davison, G.; Callister, R.; Williamson, G.; Cooper, K.A; Gleeson M. The effect of acute pre-exercise dark chocolate consumption on plasma antioxidant status, oxidative stress and immunoendocrine responses to prolonged exercise. Eur. J. Nutr. 2012, 51, 69-79. doi: 10.1007/s00394-011-0193-4.
- Morillas-Ruiz, J.; Zafrilla, P.; Almar, M.; Cuevas, M.J.; López, F.J.; Abellán, P.; Villegas, J.A.; González-Gallego, J. The effects of an antioxidant-supplemented beverage on exercise-induced oxidative stress: results from a placebo-controlled double-blind study in cyclists. Eur. J. Appl. Physiol. 2005, 95, 543-9. doi: 10.1007/s00421-005-0017-4.
- Xie, L.; Lee, S.G.; Vance, T.M.; Wang, Y.; Kim, B.; Lee, J.Y.; Chun, O.K.; Bolling, B.W. Bioavailability of anthocyanins and colonic polyphenol metabolites following consumption of aronia berry extract. Food Chem. 2016, 211, 860-8. doi: 10.1016/j.foodchem.2016.05.122.
- Howatson, G.; McHugh, M.P.; Hill, J.A.; Brouner, J.; Jewell, A.P.; van Someren, K.A.; Shave, R.E.; Howatson, S.A. Influence of tart cherry juice on indices of recovery following marathon running. Scand. J. Med. Sci. Sports. 2010, 20, 843-52. doi: 10.1111/j.1600-0838.2009.01005.x.
- Dimitriou, L.; Hill, J.A.; Jehnali, A.; Dunbar, J.; Brouner, J.; McHugh, M.P.; Howatson, G. Influence of a montmorency cherry juice blend on indices of exercise-induced stress and upper respiratory tract symptoms following marathon running--a pilot investigation. J. Int. Soc. Sports Nutr. 2015, 12, 22. doi: 10.1186/s12970-015-0085-8.
2) How blood samples were collected and how much quantity of samples per participants were collected.
Information was provided in the 2.2 section.
lines 150-153 After an overnight fast before breakfast, a venous blood sample was collected before (T0), and 15 min (T1), 1 h (T2), and 24 h (T3) after each race into serum sample tubes. One tube was collected per participant at each time-point, a total of four tubes per race. Prior to each blood sampling, participants rested for at least 10 min.
3) How placebo juice was made without polyphenols. Brief information must be made.
Information on placebo juice was added in the M&M section.
lines 142-146 The placebo juice was originally developed as a control beverage to be used in intervention studies with aronia juice. More precisely, three formulations were prepared by mixing various nutrients, matching the nutrient composition of aronia juice, in water, with the addition of artificial colors and flavors. As previously published, the similarity of these formulations to aronia juice was assessed by six food panelists so that formulation was chosen
4) Whether blood was drawn from the same participants at different time points, information must be included.
Information was included in 2.2 section.
lines 150-153 After an overnight fast before breakfast, a venous blood sample was collected before (T0), and 15 min (T1), 1 h (T2), and 24 h (T3) after each race into serum sample tubes. One tube was collected per participant at each time-point, a total of four tubes per race. Prior to each blood sampling, participants rested for at least 10 min.
5) In this study assays were conducted in duplicate. However, it is extremely essential to conduct each experiment in triplicate.
All assays were conducted in triplicate as it was stated in section 2.6 Statistical analysis except determination of protein fractions by electrophoresis that was performed in duplicate. Determination of serum protein profiles is a standard biochemical analysis that is routinely performed in clinical laboratories in monoplicate. So, we considered that it would be acceptable to perform the assay in duplicate. Furthermore, it was justified by obtaining a low CV value (< 5%) for the analysis.
6) While determination of serum protein profiles, by which method protein content in the sample was estimated, why native gel and why 9 % PAA gel was used. What is PAA. Incorporate information along with references.
A more detailed description considering the electrophoretic analysis of protein profiles was added in Section 2.4. in line with your suggestions.
lines 166-173 Native polyacrylamide gel electrophoresis (PAGE) was performed according to the manufacturer’s recommendations using a Hoefer SE 260 electrophoretic unit (San Francisco, CA, USA) for the separation of five main serum protein fractions (gamma, beta, alpha2, alpha1 and HSA) [37]. Before electrophoresis, the concentration of proteins in the serum was determined by the biuret method [35]. The experiment was performed in duplicate. In the first experiment 5 µg of serum proteins was applied per line, while in the second 7 µg of proteins was applied on 9 % polyacrylamide gel for the better separation of single serum protein fractions.
New reference number 37 was also added.
- O'Connell, T.X.; Horita, T.J.; Kasravi, B. Understanding and interpreting serum protein electrophoresis. Am. Fam. Physician. 2005, 71, 105-12. PMID: 15663032.
7) Method for determination of urea, creatinine and uric acid must be incorporated along with references.
Appropriate information on the methods for the determination of urea, creatinine, and uric acid are added in section 2.5.
lines 179-190 Briefly, creatinine and uric acid were determined by enzymatic colorimetric methods. Creatinine was converted into glycine, formaldehyde and hydrogen peroxide using creatininase, creatinase, and sarcosine oxidase, while uric acid was converted into allantoin and hydrogen peroxide using uricase. In the next step, the liberated hydrogen peroxide was converted into quinone-imine chromogen by action of peroxidase, whose color intensity was directly proportional to the creatinine or uric acid concentration (Roche Diagnostics, Basel, Switzerland). Urea was measured by the kinetic enzymatic assay in which urea was hydrolyzed by urease into carbonate and ammonium. In the next step, these products were converted into L-glutamate in the presence of glutamate dehydrogenase and coenzyme NADH. The rate of decrease in the concentration of NADH was directly proportional to the concentration of urea in the sample [38].
- Sampson, E.J.; Baird, M.A.; Burtis, C.A.; Smith, E.M.; Witte, D.L.; Bayse, D.D. A coupled-enzyme equilibrium method for measuring urea in serum: optimization and evaluation of the AACC study group on urea candidate reference method. Clin. Chem. 1980, 26, 816-26. PMID: 7379302.
Results:
1) The importance of each experiment conducted, its analysis etc must be described in the results section.
Thank you for your valuable suggestion, the corrections are introduced in the text and marked as red letters.
2) The values in Table 3 are not properly formatted.
We corrected that and appropriately formatted other Tables, too.
3) Relevant changes in protein fractions in all groups must be mentioned.
The suggestion is accepted, thank You.
lines 271-287 According to the results presented in Table 3, the abundances of γ and β obtained at time point T1 were almost unchanged compared to the baseline values, while the trends of changes in the abundances of α2, α1 and HSA were different in both groups. At time points T2 and T3, an increase in the percentage of the γ fraction was observed in both groups compared to baseline values, but this was only statistically significant (pË‚0.05) in the placebo group. Looking at the non-statistically significant changes in T2 and T3, α1 fraction abundance decreases in the placebo group in T3. In addition, the difference in mean values between T0 and T3 for the HSA fraction is greater in the placebo group (1.4 g/L) than in the chokeberry group (0.8 g/L) (p>0.05).
According to these results, HSA and α1 could contribute to a significant decrease in total serum proteins 24 h after the race in the placebo group (Table 2). In addition, the decrease in these protein fractions is probably jointly responsible for the decrease in total proteins 24 h and 7 days after the first race (Figure 2). The decrease in total proteins is likely due to proteinuria, as both total protein levels and protein profiles were altered.
In addition, we tested whether the observed slight changes in protein fractions lead to an increase or decrease in serum γ, β2, α2, α1 and HSA protein levels expressed in g/L (results not shown), but there were no changes that reached the significance level.
4) The sample size should be large enough to conclude the results appropriately.
We have agreed with the reviewer that 10 participants included in the study was not enough for strong statistical power and a higher number of participants was needed for that. In this pilot study, we tried to investigate how long-term moderate PA (runners who practice running 40 km per week) together with the simulated half-marathon races influence serum protein profiles. The number of participants from the Belgrade running club who fulfilled this requirement and wanted to participate in the study was limited to 10. We know that this pilot study does not have strong statistical power, but can offer the trend of changes in serum protein profiles. This study can be important because it offers a foundation for further studies in which a higher number of recreational participants will be included. The results of this study may be important, because they point out the possible negative effect of moderate or intensive long-term PA on the health of recreational runners, especially on those who are not under the control of professional nutritionists, trainers, and doctors or may not consume recommended quantities of nutrients or supplements which help them to recover from the negative effects of PA.
Discussion:
1) Discussion not written appropriately
2) Results needed to be analyzed, interpreted sufficiently and discussed for appropriate
conclusion.
3) All physiological parameters must be discussed.
All suggestions were accepted and the discussion section was carefully restructured and the missing parts were added. We hope that induced changes significantly improved the discussion section quality.
4) It is written in line 274) “….proteins with LMW such as HSA …”, here what is Mol wt of HSA? Please Mention
The molecular weight of HSA is reported to be 66700 Da and it was added in text along with the relevant reference.
- Lee, P.; Wu, X. Review: modifications of human serum albumin and their binding effect. Curr. Pharm. Des. 2015, 21, 1862-5. doi: 10.2174/1381612821666150302115025.
5) Why specifically α1, α2, γ-globulins levels of participants measured in this study? It must have been discussed along with reference literature.
The content of protein fractions α1, α2, ß, γ-globulins and HSA was determined as they represent the main protein fractions in serum after separation by electrophoresis. The relevant reference (number 37, O'Connell et al, 2005) was added in the text of the revised version of the manuscript (lines 260-261 as well as in method and material section 2.4.).
lines 260-261 It is known that non-albumin proteins can be separated into four different fractions by native electrophoresis: alpha1 (α1), alpha2 (α2), beta (ß) and gamma (γ) [37].
- O'Connell, T.X.; Horita, T.J.; Kasravi, B. Understanding and interpreting serum protein electrophoresis. Am. Fam. Physician. 2005, 71, 105-12. PMID: 15663032.
6) It is written that (line 316) “…. proteinuria and in-adequate protein intake could show a more pronounced effect on some non-albumin proteins than on HAS”, here please incorporate few non-albumin proteins' name and their references should have been mentioned.
The names of several low molecular proteins were added.
7) Reference for the line 337 “HSA is the major transport … of great importance” must be mentioned
The discussion section was carefully rewritten.
Conclusion
Conclusion of study has not been drawn appropriately. Several gaps are there. For ex what is the minimum time required for recovery and symptoms of those who are continuously doing PA? what does Acute aronia juice consumption means, Case study etc. must also discussed.
The conclusion section was carefully rewritten.
Others
Full form of all abbreviations (PAA, CBB, LDH, CK,LMW) must be mentioned when appear 1st time in the manuscript.
The abbreviations were checked.
Comment 2) Author must check grammatical mistake throughout the manuscript
Response: The English language was checked by a native English speaker.
Reviewer 2
Comments and Suggestions for Authors
- L24-25 – Abstract: “The significant changes in urea, creatinine and uric acid levels were noticed at 24 selected time points.” In what groups? Need to specify.
Response: Thank You for a suggestion we have specified that changes were noticed in both groups (marked green).
- Overall, the paper needs to be proofread to improve English grammar and readability. For example, L34 – “It is defined as a “PA…” What is It? Comes across vague.
Response: The English editing was done by a native English speaker.
- L100-102 – The goal of the study stated is confusing. If the aronia juice is the treatment, then “effect of consuming aronia juice Vs placebo in people performing PA” should be the focus of the study. Currently it states, “ The main goal of this pilot single-blind crossover placebo-controlled study was to examine the effect of repetitive PA with or without acute aronia juice supplementation on serum protein profiles of recreational runners running 40 km each week in selected time points before and after simulated half-marathon races”. The term repetitive PA is confusing. How was repetitive PA measured? Is that a variable? The sentence is too long. As an example, goals could be written - “The primary goal of this pilot single-blind crossover placebo-controlled study was to examine the effect of aronia juice supplementation versus placebo on serum protein profiles in male recreational runners at selected time points before and after simulated half-marathon races” This can be followed by listing other variables or secondary variables that was measured. Avoid terms like “for the more deep insight”. “More deep” are repetitive adjectives, hence redundant. Simply state what additional parameters were measured.
Response: Thank You for a valuable suggestion the lines 100-102 was carefully rewritten and the sentence line 106-107 modified (marked green).
- Throughout the document aronia group and chokeberry group are used interchangeably. Please be consistent and identify the treatment group with just one name.
Response: We apologize for the inconsistency. The manuscript was carefully checked and term aronia is used.
- Line 254-262 – “To confirm that two intensive PA….” is very confusing. This is the first time that the study involves 2 marathons is discussed. Please briefly mention the timeline along with number of PA in the methods section.
Response: The following sentence (marked green) was added in the section 2.1, lines 123-125 to support the text lines 254-262 :
The wash-out period between two simulated half-marathon races that each participant ran was 7 days (Figure 1).
- Discussion – Is reworded well, however, some more introspect into results can strengthen the discussion section. For example, L296-299 can be moved to discussion section, and this will be a value add. Review the results and move such descriptive and conclusive statements to the discussion section
Response: Thank You for the suggestion, we replaced several sentences from results to discussion section (marked green).
Comments on the Quality of English Language
Needs grammar review. Some sentences are too long and some start vaguely.
Response: The English language was checked by a native English speaker.
This manuscript is a resubmission of an earlier submission. The following is a list of the peer review reports and author responses from that submission.
Round 1
Reviewer 1 Report
Comments and Suggestions for Authors
Journal-Healthcare (MDPI)
Article: The Potential Benefits of Acute Aronia Juice Supplementation 2 on Physical Activity Induced Alterations of the Serum Protein 3 Profiles in Recreational Runners: A Pilot Study
Comments
Introduction:
1) The source of Aronia juice, its growing condition, availability in the countries, trade name, varieties of Aronia available, variety Aronia juice available commercially, a component of Aronia etc. information must be mentioned along with references.
2)Elaborate “Proteinuria is a benign phenomenon” , hyperalbuminemia, along with references
3) What are the different markers of kidney injury. It must be mentioned along with references
Methods
1) How it was decided to provide 200 mL of aronia juice to all the participants? Similarly, how it was decided whether before/after breakfast or before/after exercise these juice will be provided to all the participants. Also whether fresh juice or commercially available juice is needed to be provided to the participants. Information must be incorporated along with references.
2) How blood samples were collected and how much quantity of samples per participants were collected.
3) How placebo juice was made without polyphenols. Brief information must be made.
4) Whether blood was drawn from the same participants at different time points, information must be included.
5) In this study assays were conducted in duplicate. However, it is extremely essential to conduct each experiment in triplicate.
6) While determination of serum protein profiles, by which method protein content in the sample was estimated, why native gel and why 9 % PAA gel was used. What is PAA. Incorporate information along with references.
7) Method for determination of urea, creatinine and uric acid must be incorporated along with references.
Results:
1) The importance of each experiment conducted, its analysis etc must be described in the results section.
2) The values in Table 3 are not properly formatted.
3) Relevant changes in protein fractions in all groups must be mentioned.
4) The sample size should be large enough to conclude the results appropriately.
Discussion:
1) Discussion not written appropriately
2) Results needed to be analysed, interpreted sufficiently and discussed for appropriate
conclusion.
3) All physiological parameters must be discussed.
4) It is written in line 274) “….proteins with LMW such as HSA …”, here what is Mol wt of
HAS? Please Mention
5) Why specifically α1, α2, γ-globulins levels of participants measured in this study? It must
have been discussed along with reference literature.
6) It is written that (line 316) “…. proteinuria and in-adequate protein intake could show a
more pronounced effect on some non-albumin proteins than on HAS”, here please incorporate few non-albumin proteins' name and their references should have been mentioned.
7) Reference for the line 337 “HSA is the major transport … of great importance” must be
mentioned
Conclusion
Conclusion of study has not been drawn appropriately. Several gaps are there. For ex what is the minimum time required for recovery and symptoms of those who are continuously doing PA? what does Acute aronia juice consumption means, Case study etc. must also discussed.
Others
Full form of all abbreviations (PAA, CBB, LDH, CK,LMW) must be mentioned when appear 1st time in the manuscript.

Essentially needs improvement.
Reviewer 2 Report
Comments and Suggestions for Authors
This study aims to understand the effect of aronia juice consumption before recreational physical activity on kidney function parameters and serum protein profiles in runners. Study offers unique insight into changes in body protein turnover that could be translated not only to recreational activities but also athletes in general.
The authors have tried to provide evidence if polyphenols from aronia juice can help protein recovery after an athletic event. Please consider the following suggestions to improve the impact of the article.
11. L32 – Introduction. Recommend including a brief definition for recreational PA along with popular activities as examples.
22. L44 – Recommend providing possible pathologic conditions. Can reword “….they can progress to pathological conditions such as…List common conditions.”
33. L68 & 70 – Be consistent with citing format throughout the document and simplify the sentence. For example, “In a recent study of Li et al. (2021) it was shown that…” can be reworded to “ A recent study reported…” Cite the paper in the end consistent with rest of the document.
44. L75-86 – This section is very vague and appears to make random assumptions/associations. Recommend major editing to reduce guesswork from the researchers. For example - L76 – “We hypothesized that recreational runners due to repetitive PA (running 40 km per week) and PA during a simulated half-marathon race may have altered protein profiles due to the presence of persistent proteinuria caused by PA.” Has this statement been tested, as the statement begins with “we hypothesized.” Please provide appropriate context for this claim and cite accordingly.
Another vague statement - L84 – If there is no conclusive data on recreational athletes’ serum protein homeostasis, then what is the basis for this experiment? Is there some evidence in athletes? If so, provide this information to create a possible similar association for people who perform recreational PA.
55. L91-95 – This statement looks at investigating L76. This objective is different from the goal of the study. It is unclear which comes first or if these 2 aims are connected. At this point both goals cannot be achieved simultaneously.
66. L138 – Please provide appropriate sample size estimation for this study. An n of 10 is seems far too less to get any statistical power.
77. L144 – Results – Results start with biochemical parameters. However subject characteristics are missing. Recommend incorporating baseline characteristics be included for the study subjects. If this has been discussed in a previous paper, then this can be discussed briefly along with appropriate citation.
88. L175 – Results here help understand the goal mentioned in L 91-95 better. Is the impact of proteinuria and dehydration on serum protein homeostasis measured between 2 groups? If so, this needs to be clarified in L91-95, and classify primary and secondary outcomes to avoid confusion.
99. L347 – 352 This section does not correspond to the primary goal of the study. Isn’t the primary goal to observe the effect of aronia juice on serum protein and kidney function parameters? In conclusion would recommend reflecting significantly on primary outcome of the study rather than detailed discussion of common outcomes after a PA.
110. Discussion – Discussion section is overly focused on impact of PA on renal function, protein homeostasis, dehydration, serum protein changes etc. There is limited discussion about the results specific to action of polyphenols in aronia juice and changes in all the parameters measured in study subjects. In fact, L244-345 mentions polyphenols or aronia juice maybe once. There is severe lack of supporting evidence on impact of any polyphenols (not just from aronia juice) on parameters measured in this study. Would recommend focusing the discussion to corroborate the results on impact of polyphenols/ acute aronia juice consumption.
111. Authors need to provide perceived study strengths and limitations.
Comments on the Quality of English LanguageThere is an average command of English language in general. Some sentences seem very vague in the introduction and discussion sections. Several sentences can reduce redundancy. Would recommend extensive proofreading the whole article and revising as needed to improve the quality.